# The Presence of Meaning in Parenthood, Perceived Social Support, and Happiness of Mothers Living in Hong Kong: A Comparative Study on Younger and Older Mothers

**DOI:** 10.3390/ijerph19052730

**Published:** 2022-02-26

**Authors:** Siu-Ming To, Lei Yang, Ming-Wai Yan

**Affiliations:** Department of Social Work, The Chinese University of Hong Kong, Hong Kong, China; leiyang@cuhk.edu.hk (L.Y.); yancathy@cuhk.edu.hk (M.-W.Y.)

**Keywords:** parental meaning, social support, happiness, mothers, young mothers

## Abstract

This study investigates whether and how mothers’ meaning of parenthood and their perceived social support interact and are associated with their happiness. Similarities and differences in these areas between younger and older mothers are also explored. A total of 1292 Chinese mothers were recruited from different Hong Kong communities; 361 were 24 years of age and younger and were categorized as younger mothers, and 931 were older than 24 years of age and were categorized as older mothers. Multiple regression analyses were performed to discern direct and interaction effects. The results show that older mothers were happier than younger mothers and indicate that parental meaning and perceived social support had a positive effect on the happiness of both younger and older mothers. Notably, while the results reveal significant interaction effects between the meaning of parenthood and perceived social support from significant others and friends on the happiness of older mothers, these effects are not found among younger mothers. This study suggests that the social support young mothers receive from their significant others and friends does not amplify the association between their meaning of parenthood and happiness. These findings provide insights into the importance of synergizing social support and meaning-making among younger mothers.

## 1. Introduction

Motherhood is a significant part of many women’s lives. Especially in societies where traditional gender roles persist, being a mother is regarded as an essential role and a regular duty within a woman’s life [1]. Many expect motherhood to be associated with a certain level of happiness, especially in the East Asian societies where having children is still regarded as a normative behavior [2]. In the transition to motherhood, a woman may experience the joy of being a parent, and she may also face tremendous physical, psychological, and social adaption challenges associated with pregnancy, childbirth, and child-raising. Although these can contribute to a woman’s happiness in motherhood and maternal satisfaction, the coexistence of maternal stress can also deteriorate these emotions [3].

When facing transitional life experiences and challenging tasks, various factors have been found to influence an individual’s subjective well-being and happiness, including sociodemographic variables, personality, self-appraisal, and the presence of a supportive environment [4]. Previous studies (e.g., [5,6]) demonstrated that although becoming a parent may result in a decline in personal and marital well-being, perceived meaning in the parental role may contribute to an individual’s well-being. Uchida et al. [7] further documented that social factors such as social support, social relationships, and the fulfillment of relational commitments are also determinants of an individual’s psychological well-being in East Asia, where close relationships and social harmony are considered the basis of happiness. Although the aforementioned research separately examined the presence of meaning and social support as they related to psychological well-being, the interaction effect of these two variables is underexplored. Such research is essential, however, especially in the Chinese cultural context, because the perceived meanings of the Chinese people is presumably influenced by their perceived social relationships, and different levels of social connections may strengthen or weaken any positive outcomes related to meaning [8]. Based on these assumptions, it is essential that we consider meaning in parenthood and perceived social support into consideration when investigating the happiness of mothers.

Although we target mothers of all ages, the literature suggests that age could be a conditional factor in associations among factors that potentially impact a mother’s happiness [9]. Specifically, when the family background, maternal preparation, and social support of younger mothers are compared to those of older mothers, younger mothers are more likely to report a lower socio-economic status, insufficient parental preparation, a lack of social support, and even poorer child outcomes [10]. There are several possible reasons for this. First, early motherhood has been found to be associated with adverse health and well-being in both mother and child [11]; young women who decide to continue with their pregnancy are more likely to experience a variety of problems, such as the increased risk of attaining lower levels of education, conflicts between career development and child-rearing, and dependency on social welfare [12]. Furthermore, even though an overall provision of support has been shown to contribute to successful motherhood, a study of young mothers confirmed that few of the participants received community support for themselves and their children because of negative social perceptions of young mothers [1]. We cannot overlook the parenting challenges faced by young mothers and the complexity of their perceived social support, all of which may influence their overall happiness. As such, this study investigated whether and in what manner meaningful parenthood and perceived social support interact and associate with mothers’ happiness and explored the similarities and differences of these effects between younger and older mothers.

### 1.1. The Presence of Meaning in Parenthood and Happiness

The presence of meaning in parenthood refers to the extent to which parents comprehend, make sense of, or see the significance of their parenting experiences [13]. As one of two constructs of meaning in parenthood, the presence of meaning in parenthood reflects the degree to which parents perceive themselves as having a purpose, mission, or overarching aim in their parenthood; this is different from the search for meaning in parenthood, which concerns the degree to which parents attempt to establish their comprehension of meaning, significance, and purpose of being a parent [14,15]. Compared with the mixed findings on the influence of search for meaning, accumulated studies conducted in diverse cultural contexts and with different samples have achieved a consensus that the presence of meaning is a stable construct and always produces positive life outcomes and psychological well-being among people already exhibiting substantial meaning in their lives (e.g., [8,15,16]).

Literature on the presence of meaning demonstrated that although the financial, physical, and emotional costs of parenthood seem to cause higher stress levels, parents who perceive a sense of meaning in their roles as a parent are more likely to experience a higher level of personal growth, greater confidence of their abilities as a parent, and increased happiness [3]. Parenting that aligns with an individual’s understanding of their purpose and valued goals can serve as a motivation and actively mobilize the necessary resources to cope with challenges and further contributes to improved parenting practices and enhanced parental psychological well-being [14,17].

Although the significance of the presence of meaning has been found in different samples, few studies have addressed the patterns and impact of perceived meaning in parenthood and the psychological well-being among young mothers. Such research is essential, however, because compared with their counterparts, young mothers are generally in a lower socio-economic status and are more likely to face significant pressures in parenthood that impose adverse influences on their well-being, such as inadequate parental preparation, insufficient family support, and multiple competing tasks during the transition from youth to adulthood [1,18,19,20]. For this reason, the perceived meaning in parenthood and happiness of young mothers needs to be investigated.

With this in mind, the first hypothesis of the present study was as follows:

**Hypothesis** **1** **(H1).***The presence of meaning in parenthood will be positively associated with the happiness of both younger and older mothers*.

### 1.2. Social Support and Happiness

Social support is generally perceived as well-intentioned actions given willingly from one person to another, which may produce a positive response and contribute to an individual’s happiness [21,22]. Brownell and Shumaker [23] also characterized social support as “an exchange of resources between at least two individuals perceived by the provider or then recipient to be intended to enhance the well-being of the recipient.” Hori and Kamo [24] concluded that the positive effect of having social support is more significant for mothers; compared to the expectation role of fathers as the primary breadwinner, mothers are expected to shoulder the heavy tasks associated with child-rearing. As such, receiving social support related to parenting work, such as helping to care for children or providing financial support, can relieve the pressures of childcare and child-rearing and improve the psychological well-being of mothers [25].

There is evidence that social support may produce beneficial effects on mothers’ parenting experiences and well-being, especially young mothers. As has already been stated, compared with their counterparts, young mothers typically face more pressures in parenthood and receive less social support, which causes them to eagerly seek social support to compensate for this lack [10]. Moreover, frequent meetings with a supportive family member or peer lead to closer relationships and results in a supportive environment that enables the creation of special bonds and cooperation during stressful parenting experiences, which may engender positive psychological well-being [18]. It is, therefore, understandable in addition to releasing parents from stressful parenting tasks, social support may also create supportive relationships to mothers and thereby influence their happiness.

Based on this consideration, the second hypothesis of the present study was as follows:

**Hypothesis** **2** **(H2).**
*Perceived social support will be positively associated with the happiness of both younger and older mothers.*


### 1.3. Amplifying Effect of Social Support on Meaning in Parenthood

Apart from the direct influence of social support on the association between social support and psychological well-being, this study examined the interaction effect between the presence of meaning and social support on mothers’ happiness. Baumeister and Leary [26] argued that meaning-making arises from social relationships, which means that close social connections and a supportive environment will consolidate and even strengthen the positive influences of the meaning an individual has already perceived. As was already mentioned, there is a relational component of personal meaning-making in the Chinese cultural context; in this setting, when social support allows a closer social connection and provides a more supportive environment, the effect of perceived meaning is magnified. Moreover, because traditional Chinese culture has always emphasized children as the core of the family, parenting and parental practices are not only dominated by parents, but also heavily influenced by family members and societal expectations [27]. In this way, a synergistic effect may exist between perceived meaning in parenthood and perceived social support from family members and other significant people in a mother’s life.

It should be noted that compared to older mothers, young mothers may pay more attention to their own evaluation and self-fulfillment, which would render the meaning perceived by younger generations less susceptible to influences from external environments [17,28]. The younger generations of Chinese females grew up in a society wherein they enjoyed better socioeconomic security and adopted post-materialistic values, leading them to place a greater emphasis on individualism, exercise independence, and test new identities, compared to the older generations [29,30]. Furthermore, the childrearing experiences of young mothers may not be easily understood by their families and social circles, and they are more likely to face social exclusion [19,31]; this may challenge young mothers’ scaffolding and construction of parental meaning through social support.

For this reason, the third hypothesis for the present study is as follows:

**Hypothesis** **3** **(H3).**
*Although social support will amplify the positive association between perceived meaning in parenthood and mothers’ happiness, the interaction effect in young mothers will be weaker than that in older mothers.*


Thompson and Heller [32] asserted that different types of social support could have different effects on individual psychological well-being. Watts et al. [1] concluded that the most important social support for mothers was from family members—especially the mother’s parents—and the mother’s partner. Similarly, Mallette et al. [33] conducted a correlational study on new mothers’ subjective well-being and found that social support from the mother’s mother and the children’s father contributed to positive self-perception in relation to parenting tasks. Although friends and peers play an essential role in emotionally supporting mothers, especially new mothers, the role of peer support in motherhood is an area of research that has received significantly less attention than the effect of family and partner support [1]. Nitz et al. [34] found that friends were the second most frequently identified provider and source of support, and Dellman-Jenkins et al. [35] observed that friends played a key role when mothers needed someone with whom they could discuss daily activities and elicit emotional support. Given these findings, we will divide social support into three specific aspects—family, friends, and one’s significant other—and investigate the different moderating influences of social support received from these three sources.

## 2. Materials and Methods

### 2.1. Participants and Procedures

We used a dataset derived from a cross-sectional project co-organized with the Hong Kong Young Women’s Christian Association, which investigated young mothers’ parenting experience and life development [36]. Prior to initiating the research, we obtained ethical approval from Survey and Behavior Research Ethics Committee of The Chinese University of Hong Kong, with which the researchers are affiliated. As it relates to the accessibility of the research participants, we sent invitation letters to collaborating childcare centers, kindergartens, social service organizations, and non-government organizations to solicit their assistance in recruiting mothers to participate in this study. At the beginning of the assessment, the principles of voluntary participation, free withdrawal, and the guarantee of anonymity were clearly explained to every participant.

A total of 1348 Chinese mothers from different neighborhoods or communities in Hong Kong were recruited from collaborating organizations to complete a survey. Because mothers’ age is a focal variable in the current study, 56 mothers who did not report their age were excluded. The remaining 1292 respondents were included in this study.

Although there is no consensus on the definition of “young people,” the 15–24 age range is a generally accepted criterion to define young people [37]. During this period, individuals may experience a life transition from the dependence of childhood to the independence of adulthood [37]. For this reason, this definition was adopted in the present study to divide the participants into two different age groups. Of the 1292 mothers, 361 were aged 16–24 and categorized as young mothers, and the remaining 931 were aged 25–56 and categorized as older mothers; the mean ages of the young mothers and older mothers were 22.12 (SD = 1.780) and 35.38 (SD = 4.806), respectively.

### 2.2. Instruments

#### 2.2.1. Presence of Meaning in Parenthood

The Meaning in Parenthood questionnaire (MPQ) was developed based on the original version of the Meaning in Life Questionnaire (MLQ) (Steger et al., 2006), then was modified and validated among Chinese parents [36]. The MPQ retained the same 10-item contents as the original MLQ: five items on the presence of meaning in parenthood (MPQ-Presence) (e.g., “As a parent, I have a clear purpose”) and five items on the search for meaning in parenthood (MPQ-Search) (e.g., “I am looking for something that makes my parenthood meaningful”). In this study, we adopted the 5-item MPQ-Presence to measure the mothers’ perceived meaning in parenthood; this scale demonstrated satisfactory internal consistency (Cronbach’s alpha = 0.764) in our sample.

#### 2.2.2. Social Support

Social support was measured using the Multidimensional Scale of Perceived Social Support (MSPSS). The original version of the MSPSS was developed by Zimet et al. [10] to subjectively assess perceptions of social support adequacy from three specific sources: family, friends, and one’s significant other. The MSPSS consists of 12 items, with four items devoted to each source of social support. Participants rate each item on a 6-point Likert-type scale (i.e., 1 = Strongly Disagree, 6 = Strongly Agree) and received a total score ranging from 5 to 30, with higher total scores indicating higher levels of perceived social support. In this study, we adopted the Chinese version of this scale (MSPSS-C), which was translated by Chou [38] using a sample of Chinese youth from Hong Kong. The reliability coefficients of the Family, Friends, and Significant Other subscales were 0.903, 0.908, and 0.921, respectively.

Prior to data analyses, we examined whether the three-subscale structure of the MSPSS C (i.e., Family, Friends, and Significant Other) was replicable on the mother sample by employing a confirmatory factor analysis. We used the comparative fit index (CFI), the Tucker–Lewis Index (TLI), and the root mean square error of approximation (RMSEA) to evaluate the fit of the model. The CFI and TLI range from 0 to 1, with 0 indicating no fit and 1 indicating a perfect fit and values close to 0.95 indicating a well-fitting model [39]; the RMSEA varies from 0 to 1, with values between 0.05 and 0.08 indicating a close fit [40]; the goodness-of-fit indices of the three-subscale model were found to be acceptable (i.e., CFI = 0.98; TLI = 0.97; RMSEA = 0.075).

#### 2.2.3. Happiness

Happiness was assessed using the 6-Item Short Depression–Happiness Scale (Joseph et al., 2004). The 6-Item Short Depression–Happiness Scale was developed from the 25-Item Depression–Happiness Scale, which conceptualized happiness as a continuum of depression to happiness, rather than a unipolar measure of the absence of depression, and also considered the presence of positive thoughts and feelings [41,42]. A higher score indicates more frequent positive thoughts and feelings and less frequent negative thoughts and feelings. This study employed the Chinese version of this scale to measure the mothers’ subjective evaluation of their happiness [43]; in this sample, the coefficient alpha was 0.853.

#### 2.2.4. Control Variables

Since the mothers’ level of happiness is potentially influenced by their education and career, relationship status, and level of income, we controlled several demographic variables in our analysis, including education level, employment status, marital status, and family income. Furthermore, 8.4% of the mothers who were categorized as older mothers at the time of the survey had given birth to their first child when they were younger than 24; since becoming a mother at a younger age could have a carryover influence on a woman’s social identity at an older age, the age of the mothers’ first childbirth was considered as a control variable in our analyses [44].

### 2.3. Data Analysis

To understand the differences in demographic backgrounds between younger and older mother groups, we examined the frequencies of the demographic variables and conducted independent sample *t* tests or chi square tests on the demographic variables. Independent sample *t* tests were conducted to determine whether younger and older mothers, on average, differed in their level of happiness, the presence of meaning in motherhood, and their perceived social support from the three sources. We conducted Pearson’s correlation analyses on all relevant variables in the younger and older mother samples to gain a preliminary understanding of associations of the target and control variables with happiness. On the basis that H1 (i.e., the presence of meaning in motherhood is associated with happiness) and H2 (i.e., perceived social support from family, friends and one’s significant other is associated with happiness) were primarily supported, we investigated whether perceived social support from each of the three sources moderated the association between the presence of meaning in motherhood and happiness for younger and older mothers, respectively (i.e., H3).

To achieve this, we conducted three hierarchical regression analyses on the younger and older mother groups, with happiness as the dependent variable, each of the three subscales of perceived social support as the moderator, and the presence of meaning in motherhood as the independent variable. To reduce multicollinearity, we first computed Z scores for perceived social support (PSS) and the presence of meaning in motherhood (PM), then multiplied the scores to create the interaction term [45]. In the hierarchical regression analyses model, the control variables were entered as a block in Step 1; the presence of meaning in motherhood was included in Step 2; perceived social support from one of the sources was included in Step 3; and the interaction term (i.e., PSS × PM) was included in Step 4. We then examined the change in *R*^2^ from Steps 1 through 4 for significant effects of PM and PSS on happiness, respectively, and the interaction effect of PSS × PM on happiness. To visually depict the direction of interaction in the cases where significant interactions were detected in the regression models, the interactions were plotted at the high and low levels of perceived social support, which were defined by whether the concerned source of social support scored in the “available” range.

## 3. Results

Table 1 delineates significant differences between the two groups of mothers in the demographic variables that were controlled in the analyses (i.e., education level, marital status, employment status, family income, and mother’s age at first childbirth). Compared to the older mothers, a statistically higher proportion of younger mothers were not married (χ^2^ = 209.90, *p* < 0.001), unemployed (χ^2^ = 127.50, *p* < 0.001), subsisted on a lower income (χ^2^ = 136.64, *p* < 0.001), and attained a lower level of education (χ^2^ = 96.20, *p* < 0.001). On average, the younger and older mothers reported having 1.29 (SD = 0.526) and 1.66 (SD = 0.663) children, respectively. The mean age of the first childbirth of the younger and older mothers were 20.3 and 31.2, respectively. The mean household sizes of the younger and older mothers were 4.49 (SD = 1.549) and 4.13 (SD = 1.188), respectively.

The results of the independent sample *t* test, which are presented in Table 2, revealed that older mothers (M = 4.39, SD = 0.86) were significantly happier (*t* = −6.636, *p* < 0.001) than younger mothers (M = 3.99, SD = 1.00); older mothers (M = 4.56, SD = 0.95) perceived greater family social support than younger mothers did (M = 4.35, SD = 1.22, *t* = −2.864, *p* = 0.004). Younger and older mothers did not score differently on the Friends and Significant Other subscales of perceived social support or for presence of meaning in motherhood.

The zero-order correlations for older mothers presented in Table 3 reveal that the control variables and the target independent variables—PM and PSS from family, friends and significant other—were all positively associated with happiness. Other than the age of first childbirth and employment status, all the control variables and target independent variables for the young mothers were significantly associated with happiness (see Table 4). Notably, the presence of meaning in motherhood was also positively associated with perceived social support from family, friends and significant other within both groups of mothers.

To further investigate the inter-relationships among these variables, hierarchical regression analyses were conducted. The regression results in Table 5, Table 6 and Table 7 show that after the block of demographic variables was controlled for, the association between happiness and the presence of meaning in motherhood remained significant at the *p* < 0.001 level; this supports Hypothesis 1 and signifies that both younger and older mothers who found meaning in parenthood tended to be happier. Additionally, the association between happiness and the perceived social support from all three sources remained significant at *p* < 0.001 level, even when the influence of the presence of meaning in motherhood was ruled out; this supports Hypothesis 2 and indicates that younger and older mothers who perceive greater social support from each of the three sources tended to be happier than mothers who perceived less support from any of these sources.

Regarding Hypothesis 3, of the six regression analyses conducted, the primary effects of the older mothers’ perceived social support from friends and their significant other were qualified by significant interaction effects (friends: *β* = 0.061, *p* = 0.017; significant other: *β* = 0.067, *p* = 0.008), even though significant interactions were not found for other combinations. These interactions are graphed in Figure 1 and Figure 2, and show steeper slopes between the presence of meaning in motherhood and happiness for older mothers who perceived greater support from their friends and their significant other than those who perceived lesser support. Notably, among the older mothers, those with a similar level of presence of meaning in motherhood who perceived a higher level of social support from friends and significant others tended to be happier. Even though older mothers who perceived social support from their friends and significant other amplified the positive association between the presence of meaning in motherhood and happiness, however, these amplification effects were not observed among younger mothers. The higher total *R*^2^ from all three happiness prediction equations for older mothers presented in Table 5, Table 6 and Table 7 suggest that, on the whole, the variables in the regression analyses can explain happiness in older mothers to a greater extent than the three sources of social support.

## 4. Discussion

Due to limited knowledge on the potential synergistic effect of meaning in parenthood and social support on the well-being of parents, this study examined the effects of the presence of meaning in parenthood and perceived social support on the happiness of younger and older mothers; the possible similarities and differences of these effects between the two groups of mothers were also explored. The results indicated that the presence of meaning in parenthood and perceived social support both have a positive effect on the happiness of younger and older mothers. These findings seem to confirm the significant effects of the presence of meaning in parenthood and perceived social support on mothers’ well-being.

The positive primary effect of the presence of meaning in motherhood on happiness for both groups of mothers in the present study is implicative of the mothers’ personal growth through the manifestation of a presence of meaning. Even though previous studies [46,47] have shown that younger mothers may encounter more challenges when adapting to their new identity than older mothers, our findings indicate that when a mother finds a significant sense of meaning and personal growth in motherhood, they consider mothering to be a joyful experience, regardless of their age. These quantitative findings echo the findings of a qualitative longitudinal study undertaken by Wenham [48] that an enhancement of self with new identities and responsibilities can be found among young mothers, which provide them with a new-found purpose. According to the current study findings, however, the considerable change in effect size (i.e., *R*^2^) in Step 2 of the regression analyses seems to suggest that older mothers may be better able to make meanings and thereby generate positive feelings in motherhood; this may be associated with the reflexivity that is required for the richer life and maternal experiences of older mothers [49].

Our findings further reveal that social support influences both older and younger mothers’ happiness. Specifically, all three sources of perceived social support exert positive effects on the happiness of both older and younger mothers, even after the effect of the presence of meaning in motherhood is controlled for, which demonstrates that perceiving interpersonal support is salient to the happiness of both groups of mothers. Furthermore, based on the altered effect size (i.e., *R*^2^) in the regression analyses (see Table 5, Table 6 and Table 7), it can be deduced that the relative contribution of perceived social support from different sources for older mothers are family, significant other, then friends; for younger mothers, the order is significant other, family, then friends. It is not surprising that friends ranked the lowest for both groups of mothers, because friends are conceptually more general, compared to family and one’s significant other, and their support in relation to tangible parental support is expected to be supplementary in nature, while one’s family and significant other are expected to be more reliable and may be responsible for providing essential support [50,51]. Considering that support from one’s significant other contributes to the happiness of young mothers to a greater extent than family (see Table 5 and Table 7) and the significantly lower levels of social support perceived by younger mothers, compared to older mothers (see Table 2), it could be speculated that even though family support enhances the happiness of younger mothers, this support may not be as stable and strong as support from their significant other.

This study also examined whether different sources of social support could moderate the association between the presence of meaning in parenthood and happiness among younger and older mothers. The results indicate that social support from friends and significant others only serves as an amplifier that interacts with the presence of meaning in motherhood and affects happiness of older mothers; this suggests that older mothers’ friends and significant other are able to help them process and consolidate the meaning of motherhood. For possible reasons that will be discussed, the friends and significant others of younger mothers were not found to interact with these mothers’ meaning in parenthood.

To understand these findings, it should be noted that the older mothers in this study were in the age range in which parenthood is the norm. Older mothers were able to engage in study and work prior to becoming mothers, and they were more likely to have peers who were parents or considering parenthood and could, therefore, understand and empathize with the older mothers’ struggles. It is also possible that people in an older mother’s social network have the life experience and empathy for special family contexts, such as single-parenting, nurturing children with special needs, intergenerational conflicts, and low-income [52,53,54,55]. The present study findings seem to indicate that older mothers’ attempts to process and re-evaluate parental experiences during meaning-making can be enhanced by the support from friends and significant others. Conversations with their peers and significant other may provide older mothers with additional perspectives for resolving a parenting situation, which facilitates meaning-making, and as their concerns about and efforts in motherhood are acknowledged, they are encouraged to view motherhood in a positive manner, despite challenges they are facing.

In contrast, this amplification effect of perceived social support from friends and a significant other was not observed in our sample of younger mothers. This could be due to the perceived lack of available social support from friends and a significant other that is specific to the need of young mothers’ meaning in motherhood. For example, a young mother who perceives general high social support from her friends may believe that her friends would be available to listen to her and share with her personal feelings; since a young mother’s friends are more than likely not yet mothers, however, the young mother would not expect her friends—even those who are generally supportive—to understand her parenting difficulties and concerns, or to listen to her daily childrearing concerns and offer to help care for her children. Since the meaning of motherhood is rooted in the reality of daily life, the subjective motherhood experiences of younger mothers can be expected to differ from what their friends and significant others speculate they will be [56]. Vik and DeGroot [57] found that mothers feel the safest disclosing their parenting challenges to other mothers in similar situations. In the case of young mothers, the targets of disclosures about parenting challenges may not be their non-parent peers who can only provide general social support.

The phenomenon among young mothers of not expecting their friends to understand them may also be partially explained by the persistence of equality matching as a core defining criterion in peer relationships across an entire lifespan [58,59]. Because equality matching implies that peers give and take the same amount of affection and support, even though parenting is an important part of young mother’s life, they might avoid dominating communications with their non-parent peers with parenthood-related issues. Researchers have also discovered that mothers choose to conceal or omit certain parts of parenthood because they do not want themselves, their child, their spouse, and/or their parenting to be judged [57]. Young mothers with few peers in similar situations are susceptible to being ashamed of their emotions and may, therefore, conceal their struggles to an even greater extent; when this is the case, their construction of parental meaning is not enhanced by the support from their peers.

Even though young mothers may make new acquaintances to compensate for the loss of original social contact because of motherhood, researchers have found that young mothers do not necessarily feel supported in the parenthood domain by other mothers. To avoid being judged, young mothers tend to subconsciously employ an “othering” strategy to distinguish themselves from the assumed “bad mothers” [57,60,61]. It is possible that younger mothers avoid sharing their most genuine and vulnerable parenting concerns to avoid being judged; when this happens, even though this type of interaction serves a socialization purpose, it does not enhance the effect of meaningfulness on young mothers’ happiness. Although older mothers also face the problem of other mothers in their social group being judgmental and unsupportive, this problem seems to be more salient to young mothers with lower self-esteem, who are already more likely to face social stigma [31,57].

Although the findings of the present study demonstrate that the social support from a mother’s family, friends, and significant others are all important to her happiness, this support does not amplify the association between their parental meaning and happiness in young mothers. We speculate that this may be due to support that does not echo with the core needs, perspectives, and meaning that are unique to young mothers, and as a result, young mothers may continue to feel lonely and isolated on their path of meaning-making in motherhood.

### 4.1. Implications for Practice

To help younger mothers acquire adequate social support that would consolidate their parental meaning, the young mothers themselves may need the space and opportunity to engage in an in-depth exploration of their subjective motherhood experiences. A comprehensive needs assessment could be conducted to uncover the unique concerns and issues faced by young mothers when they are juggling different roles, about which they would normally not have an opportunity to discuss. Social service providers could assist in the setup of non-judgmental real-life and online mutual support groups for younger mothers; in addition to being able to connect for mutual tangible and emotional support that they otherwise lack, platforms that encourage narrative sharing and critical reflections of prevailing ideologies would also provide younger mothers with a safe opportunity to share their parenting views and life episodes that could go against the current dominant discourses [62]. Through continuous dialogues with other young mothers in a friendly and non-judgmental atmosphere, these women would not only be able to scaffold and reflect on their inner voices, but they could also clarify their perspectives on parenthood [63]. These mothers could then actively integrate their unique lived experiences and construct and deepen their meaning in parenthood by sharing with other young mothers [64]. Through the mutual support group, young mothers could concurrently address and reappraise their stressful parenting experiences and reduce their parental anxiety through meaning-making.

Regardless of their effect, relationships in a young mother’s existing network are the most salient aspect of their daily functioning and experiences. In addition to helping young mothers gather support among themselves, social workers could also discuss possible communication strategies with young mothers so these women can effectively assert their perspectives in their interactions with friends, family, and their significant other. Social service providers could also include a young mother’s family members, friends, and significant other in sharing sessions that cover a specific theme, such as what they think about raising children under the dominant discourses, to provide opportunities for young mothers to share their unique perspectives of motherhood and to feel better understood. In this way, instead of dwelling on the personal defects and special needs of a young mother, her thoughtful parental decisions, personal strengths, and lifelong potential are highlighted. When the people in her social network understand her parenting meanings, a young mother can safely explore her parental concerns and share her parental struggles and satisfactions, even if she is considered to be a minority group in the society.

### 4.2. Research Implications

Obviously, it would be a pity if young mothers who harbor a sense of maternal meaning and significant positive potential are unable to be empowered because of scarce motherhood-specific social support. To benefit younger mothers’ parental functioning, aspects of perceived social support that are specific to young mothers’ core identity of being a mother must be studied more thoroughly.

Moreover, this study only taps at the surface of this topic, but does not delve into the sources or contents of the meaning in motherhood [36]. Future research on the perceived sources of parental meaning and sense of meaningfulness in motherhood for young mothers and the possible interaction effects thereof on maternal well-being should be undertaken. Moreover, to better understand the dynamics between happiness and a mother’s sense of meaningfulness in motherhood in specific social contexts, qualitative interviews or focus groups should be conducted to uncover the conflicting views young mothers and their supporters regarding parenthood and investigate the tendency of young mothers to disclose parenting challenges to different parties and their reasons for this.

### 4.3. Limitations

Caution is needed when interpreting the three sources of perceived social support in this study, especially “family” and “significant other.” Although it is necessary to distinguish the different sources of perceived support, it should be noted that these categorizations are not exhaustive. Even though the three-factor structure of MSPSS was applicable to the current sample, each mother may have had different representations in her mind when responding to MSPSS; for example, the item “special person with whom I can share my joy and sorrows” could be her partner, a family member, or one of her friends. It can also be expected that the “significant other” in the mind of a young mother could be, for example, neighbors and/or community agencies. To further understand the unique constitutions of “family,” “friends,” and “significant others” for the mothers prior to service provision, a detailed network listing—which was outside the scope of this study—could be conducted.

The generalizability of these findings should also be further scrutinized, because the current study was not based on randomized representative samples of younger and older mothers. Furthermore, because a self-reported survey was used in this study, anonymity and confidentiality of information were emphasized, but the responses may have been biased due to social desirability.

Furthermore, considering mothers generally experience psychological and social adaption challenges in their parenthood, a decline in their personal and marital well-being is often expected, particularly when their children are still young (e.g., [3,14]). Thus, future studies may take these negative influencing variables into consideration, such as parental stress and anxiety, and investigate their associations with maternal meaning and well-being.

## 5. Conclusions

Whereas the current findings indicate that younger mothers perceive mothering as an endeavor that is every bit as meaningful as older mothers’ perceptions, the social support they received from their family, friends, and significant other may fail to moderate the association between their sense of meaningfulness in motherhood and their happiness. This suggests there may be issues hindering younger Hong Kong mothers from enhancing their parental meaning and well-being through the facilitation and amplification of social support. These findings potentially pose significant implications on the effect that strengthening social support from the family, friends, and significant other of young mothers has on meaning-making in motherhood. Parenting interventions targeting young mothers should, therefore, be meaning-focused and support-oriented to maximize the synergistic effect of these two components.

## Figures and Tables

**Figure 1 ijerph-19-02730-f001:**
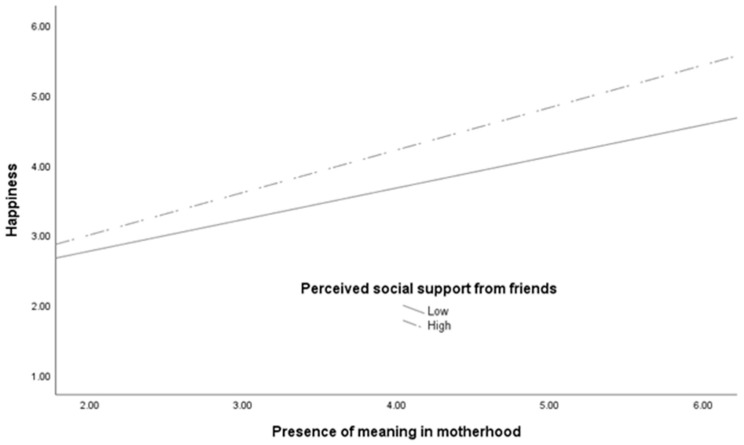
Association of older mothers’ presence of meaning in motherhood and happiness as a function of their perceived social support from friends.

**Figure 2 ijerph-19-02730-f002:**
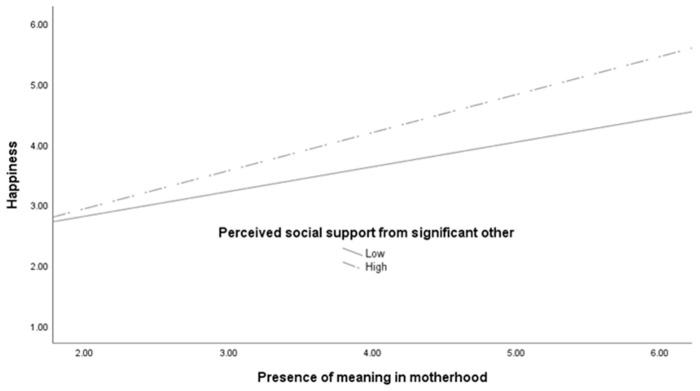
Association of older mothers’ presence of meaning in motherhood and happiness as a function of their perceived social support from significant other.

**Table 1 ijerph-19-02730-t001:** Demographics of younger (aged 24 or below) and older mothers (aged above 24).

	Younger Mothers (*n* = 361)	Older Mothers (*n* = 931)		
	*n* (%)	*n* (%)	χ^2^ or *t*	*p*
Education level			96.201	0.000
Primary to Junior secondary (F.3)	113 (31.5)	173 (19.0)		
Senior secondary (F.4-6/7) or vocational training to Post-secondary	231 (64.3)	473 (52.1)		
Bachelor’s degree or above	15 (4.2)	262 (28.9)		
Marital status			209.895	0.000
Not married	134 (37.1)	52 (5.6)		
Married	227 (62.9)	879 (94.4)		
Employment status			127.501	0.000
Not employed	260 (72.6)	334 (37.4)		
Employed	98 (27.4)	560 (62.6)		
Family income			136.640	0.000
$20,000 or below	219 (60.7)	257 (28.2)		
$20,001–$50,000	128 (35.4)	451 (49.3)		
$50,001 or above	14 (3.9)	206 (22.5)		
Age of first childbirth			−54.781	0.000
	*n* = 300	*n* = 894		
	*M* = 20.31	*M* = 31.19		

**Table 2 ijerph-19-02730-t002:** Independent sample *t* tests comparing the presence of meaning in motherhood, perceived social support, and happiness among younger mothers (*n* = 361) and older mothers (*n* = 931).

Scales	Group	Mean (SD)	*t*
PM	Younger	4.47 (0.74)	−0.378
	Older	4.49 (0.70)	
PSS from family	Younger	4.35 (1.22)	−2.864 *
	Older	4.56 (0.95)	
PSS from friends	Younger	4.37 (1.20)	−1.871
	Older	4.50 (0.92)	
PSS from significant other	Younger	4.48 (1.14)	−1.734
	Older	4.60 (0.97)	
Happiness	Younger	3.99 (1.00)	−6.636 ***
	Older	4.39 (0.86)	

Note: * *p* ≤ 0.05, *** *p* ≤ 0.001; PM = presence of meaning in motherhood; PSS = perceived social support.

**Table 3 ijerph-19-02730-t003:** Summary of means, standard deviations, and zero-order correlations among the demographic control variables, presence of meaning in motherhood, perceived social support (PSS), and happiness among older mothers (*n* = 931).

Older Mothers										
	Mean (SD)	1	2	3	4	5	6	7a	7b	7c
1. Education Level		1								
2. Marital status		0.073 *	1							
3. Employment status		0.382 ***	0.048	1						
4. Family income		0.585 ***	0.161 ***	0.353 ***	1					
5. Age of first childbirth	31.19 (4.76)	0.194 ***	0.067 **	0.131 ***	0.204 ***	1				
6. PM	4.49 (0.70)	0.106 ***	0.028	0.046	0.130 ***	0.108 **	1			
7a. PSS from family	4.56 (0.95)	0.096 **	0.137 ***	0.106 **	0.196 ***	0.074 **	0.417 ***	1		
7b. PSS from friends	4.50 (0.92)	0.110 **	0.033	0.163 ***	0.202 ***	0.046	0.393 ***	0.695 ***	1	
7c. PSS from significant other	4.60 (0.97)	0.089 *	0.094 **	0.120 ***	0.200 ***	0.059	0.409 ***	0.845 ***	0.727 ***	1
8. Happiness	4.39 (0.86)	0.215 ***	0.111 ***	0.126 ***	0.338 ***	0.101 **	0.536 ***	0.635 ***	0.560 ***	0.597 ***

Note: * *p* ≤ 0.05; ** *p* ≤ 0.005; *** *p* ≤ 0.001; PM = presence of meaning in motherhood; PSS = perceived social support.

**Table 4 ijerph-19-02730-t004:** Summary of means, standard deviations, and zero-order correlations among the demographic control variables, presence of meaning in motherhood (PM), perceived social support (PSS), and happiness among younger mothers (*n* = 361).

Younger Mothers										
	Mean (SD)	1	2	3	4	5	6	7a	7b	7c
1. Education Level		1								
2. Marital status		0.194 ***	1							
3. Employment status		0.185 ***	−0.024	1						
4. Family income		0.301 ***	0.254 ***	0.167 ***	1					
5. Age of first childbirth	20.31 (20.6)	0.264 ***	0.201 ***	0.007	0.168 **	1				
6. PM	4.47 (0.74)	−0.021	0.085	−0.052	0.166 **	−0.010	1			
7a. PSS from family	4.35 (1.22)	0.056	0.074	0.057	0.149 **	0.063	0.359 ***	1		
7b. PSS from friends	4.37 (1.20)	0.103	0.057	0.065	0.120 *	0.094	0.313 ***	0.688 ***	1	
7c. PSS from significant other	4.48 (1.14)	0.085	0.071	0.075	0.166 **	0.072	0.338 ***	0.784 ***	0.765 ***	1
8. Happiness	3.99 (1.00)	0.126 *	0.169 ***	0.047	0.159 **	0.008	0.435 ***	0.564 ***	0.454 ***	0.575 ***

Note: * *p* ≤ 0.05; ** *p* ≤ 0.005; *** *p* ≤ 0.001; PM = presence of meaning in motherhood; PSS = perceived social support.

**Table 5 ijerph-19-02730-t005:** Hierarchical regression analyses predicting happiness from presence of meaning in motherhood and perceived social support from family among older (*n* = 819) and younger mothers (*n* = 295).

Happiness						
	Older Mothers			Younger Mothers		
Predictor	∆*R*^2^	*β*	*t*	∆*R*^2^	*β*	*t*
Step 1	0.110			0.054		
Demo:					-	-
Education level		−0.004	−0.102		0.053	0.860
Employment status		0.008	0.222		0.012	0.211
Marital status		0.046	1.365		0.103	1.697
Family income		0.310	7.362 ***		0.168	2.714 **
Age of first childbirth		0.038	1.114		−0.053	−0.885
Step 2	0.243			0.191		
Demo		-	-		-	-
PM		0.499	17.459 ***		0.447	8.535 ***
Step 3	0.184			0.179		
Demo		-	-		-	-
PM		0.312	11.865 ***		0.272	5.507 ***
PSS from family		0.479	17.964 ***		0.464	9.441 ***
Step 4	0.002			0.002		
Demo		-	-		-	-
PM		0.315	11.960 ***		0.282	5.600 ***
PSS Fam		0.481	18.055 ***		0.467	9.489 ***
PM × PSS from family		0.042	1.765		0.047	1.018
Total *R*^2^	0.539			0.426		

Note: ** *p* < 0.01; *** *p* <.001; Demo = demographic (education level, marital status, employment status, family income and age of first childbirth); PM = presence of meaning in motherhood; PSS = perceived social support.

**Table 6 ijerph-19-02730-t006:** Hierarchical regression analyses predicting happiness from presence of meaning in motherhood and perceived social support from friends among older (*n* = 819) and younger mothers (*n* = 295).

Happiness						
	Older Mothers			Younger Mothers		
Predictor	∆*R*^2^	*β*	*t*	∆*R*^2^	*β*	*t*
Step 1	0.110			0.054		
Demo:		-	-		-	-
Education level		−0.004	−0.102		0.053	0.860
Employment status		0.008	0.222		0.012	0.211
Marital status		0.046	1.365		0.103	1.697
Family income		0.310	7.362 ***		0.168	2.714 **
Age of first childbirth		0.038	1.114		−0.053	−0.885
Step 2	0.243			0.191		
Demo		-	-		-	-
PM		0.499	17.459 ***		0.447	8.535 ***
Step 3	0.128			0.116		
Demo		-	-		-	-
PM		0.352	12.766 ***		0.331	6.521 ***
PSS from friends		0.397	14.174 ***		0.364	7.211 ***
Step 4	0.004			0.000		
Demo		-	-		-	-
PM		0.356	12.925 ***		0.335	6.428 ***
PSS from friends		0.404	14.396 ***		0.364	7.200 ***
PM × PSS from friends		0.061	2.396 *		0.014	0.293
Total *R*^2^	0.485			0.361		

Note: * *p* < 0.05; ** *p* < 0.01; *** *p* < 0.001; Demo = demographic (education level, marital status, employment status, family income and age of first childbirth); PM = presence of meaning in motherhood; PSS = perceived social support.

**Table 7 ijerph-19-02730-t007:** Hierarchical regression analyses predicting happiness from presence of meaning in motherhood and perceived social support from significant other among older (*n* = 819) and younger mothers (*n* = 295).

Happiness						
	Older Mothers			Younger Mothers		
Predictor	∆*R*^2^	*β*	*t*	∆*R*^2^	*β*	*t*
Step 1	0.110			0.054		
Demo		-	-		-	-
Education level		−0.004	−0.102		0.053	0.860
Employment status		0.008	0.222		0.012	0.211
Marital status		0.046	1.365		0.103	1.697
Family income		0.310	7.362 ***		0.0168	2.714 **
Age of first childbirth		0.038	1.114		−0.053	−0.885
Step 2	0.243			0.191		
Demo		-	-		-	-
PM		0.499	17.459 ***		0.447	8.535 ***
Step 3	0.153			0.181		
Demo		-	-		-	-
PM		0.331	12.217 ***		0.284	5.814 ***
PSS from SO		0.435	15.821 ***		0.464	9.529 ***
Step 4	0.004			0.000		
Demo		-	-		-	-
PM		0.330	12.197 ***		0.284	5.748 ***
PSS from SO		0.445	16.095 ***		0.464	9.508 ***
PM × PSS from SO		0.067	2.670 **		0.000	0.001
Total *R*^2^	0.510			0.426		

Note: ** *p* < 0.01; *** *p* < 0.001; Demo = demographic (education level, marital status, employment status, family income and age of first childbirth); PM = presence of meaning in motherhood; PSS = perceived social support.

## Data Availability

The datasets generated during and analyzed during the current study are not publicly available due to datasets containing information that could compromise the privacy of research participants. The data that support the findings of this study are available from the corresponding author (S.-M.T.) upon reasonable request.

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
