# Peer review of "The Presence of Meaning in Parenthood, Perceived Social Support, and Happiness of Mothers Living in Hong Kong: A Comparative Study on Younger and Older Mothers"

_ijerph, 2022, doi:10.3390/ijerph19052730_

Round 1

Reviewer 1 Report

The authors examine the problem of meaning in parenting, perceived social support, and happiness in young and older mothers living in Hong Kong. The results showed that older mothers are happier than younger mothers and confirmed that parental meaning and perceived social support have a positive impact on the happiness of both younger and older mothers. 
The work is well-documented, the study was performed on a large sample of women, and detailed statistical analyses were conducted. However, in my opinion, the results obtained were expected.
In my opinion, the weak point of the paper is the limitation of the research to East Asian societies, there is no reference to European societies, which may make it of limited interest to a larger group of researchers.

Minor revision
The work requires editorial corrections.

Reviewer 2 Report

Dear Authors,
it is a great pleasure to read this well written paper. The aim was clear and the method/material too. Results are fully explained. The issues undertaken in the work are important and bring important elements to this field of knowledge.

Please just pay attention to References, because only 16 of 63 references date 5 years or less. Could you integrate more recent sources?

Reviewer 3 Report

A shortcoming of the study is the omission of parenting stress as an important factor in the experience of parenting. Perhaps the topic of parental stress or parental burnout should be included in the theoretical considerations or discussion. It seems that stress negatively influences the level of happiness and may itself depend on the level of support and sense of meaning of parenthood. Parental stress may be an important mediator in the described pattern of variables. I submit the above for your consideration. These are not strict recommendations.
